# Which Is the Most Appropriate Cut-Off of HbA1c for Prediabetes Screening in Caucasian Youths with Overweight or Obesity?

**DOI:** 10.3390/ijerph20020928

**Published:** 2023-01-04

**Authors:** Procolo Di Bonito, Maria Rosaria Licenziati, Domenico Corica, Malgorzata Wasniewska, Anna Di Sessa, Emanuele Miraglia del Giudice, Anita Morandi, Claudio Maffeis, Maria Felicia Faienza, Enza Mozzillo, Valeria Calcaterra, Francesca Franco, Giulio Maltoni, Giuliana Valerio

**Affiliations:** 1Department of Internal Medicine, “S. Maria delle Grazie” Hospital, 80078 Pozzuoli, Italy; 2Neuro-Endocrine Diseases and Obesity Unit, Department of Neurosciences, Santobono-Pausilipon Children’s Hospital, 80139 Napoli, Italy; 3Department of Human Pathology in Adulthood and Childhood, University of Messina, 98125 Messina, Italy; 4Department of Woman, Child and of General and Specialized Surgery, University of Campania “Luigi Vanvitelli”, 80138 Napoli, Italy; 5Department of Surgery, Dentistry, Pediatrics and Gynecology, Section of Pediatric Diabetes and Metabolism, University and Azienda Ospedaliera Universitaria Integrata of Verona, 37126 Verona, Italy; 6Department of Precision and Regenerative Medicine and Ionian Area, University of Bari “Aldo Moro”, 70121 Bari, Italy; 7Section of Pediatrics, Department of Translational Medical Science, Regional Center of Pediatric Diabetes, University of Naples “Federico II”, 80131 Napoli, Italy; 8Pediatric Department, “V. Buzzi” Children’s Hospital, 20154 Milano, Italy; 9Department of Internal Medicine, University of Pavia, 27100 Pavia, Italy; 10Pediatric Department, Azienda Sanitaria Universitaria Friuli Centrale, Hospital of Udine, 33100 Udine, Italy; 11Pediatric Unit, IRCCS Azienda Ospedaliero-Universitaria di Bologna, 40138 Bologna, Italy; 12Department of Movement Sciences and Wellbeing, University of Napoli “Parthenope”, 80133 Napoli, Italy

**Keywords:** disposition index, glycated hemoglobin A1c, insulin resistance, insulin sensitivity, pediatric obesity, prediabetes

## Abstract

This cross-sectional study aimed to assess the best cut-off of HbA1c for detection of impaired fasting glucose (IFG), impaired glucose tolerance (IGT), beta-cell impairment and cardiometabolic risk (CMR) profile in overweight or obese (OW/OB) Caucasian youths. Two-hour oral glucose tolerance test was available in 1549 youths, one-hour glucose (G_60_) in 1430 youths and disposition index (DI) in 972 youths. Insulin resistance (IR) was calculated as Homeostatic Model Assessment for IR and insulin sensitivity (IS) as 1/fasting insulin. High G_60_ was defined by a value ≥ 133 mg/dL. The best cut-off of HbA1c for IFG or IGT was 5.5%. The frequency of individuals with HbA1c ≥ 5.5% was 32.5%, compared to 16.3% with HbA1c ≥ 5.7% (as proposed by the American Diabetes Association). HbA1c ≥ 5.5% showed higher sensitivity and lower specificity with respect to HbA1c ≥ 5.7% for all the abnormalities examined (IFG, IGT, high G_60_, IR, low IS, DI and CMR factors). In conclusion, this lower cut-off might represent a more appropriate screening marker of glucose dysmetabolism in youths with OW/OB. Prospective studies are needed to validate this cut-off for predicting prediabetes/diabetes in youths with OW/OB.

## 1. Introduction

Glycated hemoglobin (HbA1c) has been proposed as a useful tool for monitoring glycemic control in patients with diabetes mellitus (DM) since 1976 [1]. On the contrary, its use in the screening and diagnosis of DM and prediabetes is relatively recent. Indeed, in 2010, the American Diabetes Association (ADA) adopted the cut-off values of ≥6.5% and ≥5.7%, respectively, for the diagnosis of DM and prediabetes [2]. This choice derived from the observation that HbA1c is a more stable parameter than fasting glucose, as it reflects the average blood glucose values of the previous 120 to 180 days, and therefore, it is not affected by acute changes in glucose levels.

In recent years, the prevalence of overweightness or obesity (OW/OB) in childhood has dramatically increased [3], with the consequent high risk of developing an abnormal glucose metabolism. Therefore, in this population the screening of prediabetes/diabetes is recommended, especially in youths with at least one adjunctive risk factor [4]. With regard to the screening of prediabetes/diabetes, the ADA extended to children and adolescents the same cut-offs of HbA1c adopted in adults [4]. However, their validity in children and adolescents remains still debatable, since longitudinal studies supporting these cut-offs were available only in adults [4].

With regard to prediabetes, a number of studies evaluated several cut-offs of HbA1c from 5.4 to 5.8% in relation to the detection of impaired fasting glucose (IFG) and/or impaired glucose tolerance (IGT) [5,6,7,8,9,10]. Using a high specificity cut-off, a large number of individuals at risk of prediabetes might be excluded from a tighter management and follow-up. In line with this concern, a recent study performed in a large school-based population demonstrated that the prevalence of youths with a cut-off ≥5.7% was very low (about 2%) in young people with normal weight or with obesity [11]. For this reason, the authors suggested caution in using this cut-off for screening of prediabetes in young people.

Since most studies were performed in multi-ethnic populations, the aim of our cross-sectional study was to assess the best cut-off of HbA1c in terms of sensitivity and specificity for detection of IFG or IGT, measures of beta-cell function and abnormal CMR profile in a large sample of Caucasian youths with OW/OB.

## 2. Materials and Methods

This cross-sectional multicenter study was undertaken within the Childhood Obesity study group of the Italian Society for Pediatric Endocrinology and Diabetology (ISPED). Nine tertiary Italian centers for the care of pediatric obesity provided anthropometric and biochemical data for 1549 non-diabetic children and adolescents with OW or OB aged 5 to 18 years consecutively observed in the period June 2016 to June 2020. Other exclusion criteria were genetic causes of obesity, systemic and endocrine diseases, and use of medications affecting glucose metabolism, as elsewhere described [12].

The study was approved by the Ethics Committee of the AORN Santobono-Pausilipon, Naples, Italy (protocol code 22877/2020) and conformed to the guidelines of the European Convention of Human Rights and Biomedicine for Research in Children as elsewhere described. The study was also in accordance with the 1975 Declaration of Helsinki, revised in 2013, and informed consent was obtained from the parents or tutors of all participants.

### 2.1. Measurements

Body mass index (BMI) was calculated as weight/height^2^ (kg/m^2^) and it was subsequently transformed into standard deviation score (SDS), based upon the Italian BMI reference standards [13]. Systolic and diastolic blood pressure was measured using aneroid sphygmomanometer with cuffs of appropriate size, according to standard procedures. After 12 h of fasting, blood samples were drawn for glucose (G_0_), insulin (I_0_), HbA1c, triglycerides (TG), total cholesterol, high density lipoprotein-cholesterol (HDL-C) measurements. Oral glucose tolerance test (OGTT) was performed in the whole sample using 1.75 g/kg of glucose up to a maximum of 75 g and two-hour post-load glucose (G_120_) was measured. Data of glucose (G_30_) and insulin (I_30_) at 30′ during OGTT were available in a subsample of 972 youths, while data regarding 1-h glucose (G_60_) levels were available in 1430 individuals. Insulin resistance (IR) was calculated by homeostatic model assessment (HOMA-IR). Beta-cell function was estimated by evaluating insulin sensitivity (IS), insulinogenic index (IGI) and disposition index (DI). IS was calculated as 1/fasting insulin (I_0_) [12]; IGI was calculated as Δ(I_0_-I_30_)/Δ(G_0_-G_30_), where insulin was expressed as µU/mL and glucose as mg/dL; DI was calculated according to the following formula: IGI × 1/I_0_ [12].

Biochemical analyses were performed in the centralized laboratory of each center. HbA1c was assessed by high performance liquid chromatography. All laboratories belong to the Italian National Health System and are certified according to International Standards ISO 9000 (www.iso9000.it/ accessed on 14 November 2022), undergoing to semi-annual quality controls and inter-lab comparisons.

### 2.2. Definitions

IFG was defined by fasting glucose ≥100 mg/dL but <126 mg/dL. IGT was defined as 2 h-post-load glucose ≥140 mg/dL but <200 mg/dL [4]. High G_60_ was defined by a cut-off ≥133 mg/dL [14,15]. OW and OB were defined on the basis of the Italian BMI standards (respectively, the 75th and 95th percentiles) [13]. IR was estimated by 97th percentile of HOMA-IR distribution by age and gender in normal weight Italian children [16]. Low IS or low DI were defined by the 25th percentile of, respectively, 1/I_0_ or DI, as calculated in our sample (as elsewhere described) [12]. High TG/HDL ratio and COL/HDL ratio were defined by 75th percentile of their distribution in our sample (≥2.4 and ≥4.0, respectively). High levels of alanine aminotransferase (ALT), as surrogate marker of nonalcoholic fatty liver disease, were defined using a cut-point >25.8 IU/L in boys and 22.1 IU/L in girls [17].

### 2.3. Statistical Analysis

Data are expressed as means ± standard deviation, or proportions (%) and 95% confidence interval (CI). Given the skewed distribution of TG, HOMA-IR, I_0_ and 1/I_0_, the statistical analysis of these variables was applied after log-transformation and expressed as median and interquartile range of non-transformed values. Between-groups differences were evaluated by Student’s *t* test. Distribution of categories was evaluated by χ^2^ and, when needed, exact tests were performed using the Monte Carlo method. The relationships between IFG or IGT and HbA1c were analyzed using receiver operator curve (ROC) analysis. The area under curve (AUC) was obtained using IFG or IGT as dependent variables and HbA1c as variable of interest. The best cut-off of HbA1c was obtained using Youden’s test. AUC, sensitivity and specificity were calculated by 2 × 2 tables.

A *p* value < 0.05 was considered statistically significant. The statistical analysis was performed using the IBM SPSS Statistics, Version 20.0. Armonk, NY, USA.

## 3. Results

The whole sample was composed of 774 boys and 775 girls (mean age 11.6 years, range 5–18). Their characteristics were similar for sex and age to the subsample in whom G_60_ (716 boys and 714 girls, mean age 11.7 years,) or DI were available (494 boys and 478 girls, mean age 11.8 years). The prevalence of IFG and IGT was 10.2% and 8.0%, respectively. The AUC of HbA1c was 0.695 (95% CI 0.650–0.740) (*p* < 0.0001) with respect to IFG and 0.581 (95% CI 0.527–0.636) (*p* = 0.005) with respect to IGT. Using the Youden’s test, the best cut-off of HbA1c with respect to IFG was 5.5% (Figure 1, top panel). The same cut-off was obtained by plotting HbA1c versus IGT (Figure 1, bottom panel).

The characteristics of the sample divided by the two HbA1c cut-offs are summarized in Table 1.

The percentage of youths with HbA1c ≥ 5.5% was 32.5% vs. 16.43% with HbA1c ≥ 5.7% (*p* < 0.0001). Independently of the cut-off used, youths with either HbA1c ≥ 5.5% or ≥5.7% showed higher values of G_0_, G_60_, G_120_, HOMA-IR, TG/HDL-C, COL/HDL-C and ALT, and lower IS than the respective groups with HbA1c < 5.5% or <5.7%. No significant differences were found by comparing groups with HbA1c ≥ 5.5% to HbA1c ≥ 5.7%.

Similar differences, except for ALT, were found in the comparison between youths reclassified by HbA1c levels below or above 5.5%, after excluding the 253 individuals with HbA1c ≥ 5.7% (Table 2).

Interestingly, out of 14 youths with both IFG and IGT, 7 (50%) had HbA1c ≥ 5.5%, while none of them exhibited HbA1c ≥ 5.7%.

The performance of HbA1c ≥ 5.5 or ≥5.7% with respect to IFG, IGT, G_60_ ≥ 133 mg/dL, IR, low IS, low DI and CMR factors is summarized in Table 3. For each component of impaired glucose metabolism, beta cell dysfunction or cardiometabolic risk factor, HbA1c ≥ 5.5% showed higher sensitivity and lower specificity as compared to HbA1c ≥ 5.7%.

## 4. Discussion

The present study demonstrated that the cut-off of HbA1c ≥ 5.5% showed a better balance of sensitivity and specificity to identify individuals with IFG, IGT, high G_60_, insulin-resistance, low insulin sensitivity, low disposition index and CMR factors compared to the cut-off ≥ 5.7% proposed by the ADA. Therefore, this lower cut-off might represent a more appropriate screening marker of glucose dysmetabolism in youths with OW/OB.

In 2018, a position statement of ADA proposed the adult cut-off of HbA1c ≥ 5.7% for the screening of prediabetes in children and adolescents with OW/OB [18].

Several studies carried out in pediatric populations, which evaluated the performance of different HbA1c thresholds with respect to association with other prediabetes phenotypes, reported contrasting results. In a large sample of adolescents with OW/OB from the National Health and Nutrition Examination Surveys (NHANES 1999–2006), Lee et al. reported that a cut-off of HbA1c ≥ 5.7% showed an 11% sensitivity and 97% specificity for the identification of IFG and 5% sensitivity and 96% specificity for the identification of IGT [19]. In contrast, Tsay et al., in a multiethnic sample of 209 American youths with OW/OB, showed that HbA1c was significantly associated with IGT, while fasting glucose was not. They reported that the best cut-off was 5.5% and that it showed high sensitivity (85.7%) and poor specificity (56.9%) [20]. Similarly, in a multiethnic cross-sectional study of 1156 children with OB in the United States, Nowicka et al. showed that the best cut-off of HbA1c for IGT was 5.5% (sensitivity 67.7%, and specificity 5.5%) [7]. Baseline HbA1c strongly predicted prediabetes/diabetes in the follow-up study. In particular, adolescents with HbA1c ≥ 5.7% had greater odds of having prediabetes/diabetes after two years than their peers with HbA1c < 5.7%. Unfortunately, no information was available about the performance of cut-off points <5.7% [7]. Lastly, Poon et al. confirmed that HbA1c ≥ 5.5% was the best cut-off in identifying abnormal OGTT (prediabetes or T2DM) (sensitivity 66.7%, specificity 71%) in Chinese adolescents with OW/OB [10].

To the best of our knowledge, our study is the first to compare the association between the cut-off of HbA1c as proposed by the ADA and a lower cut-off obtained in a large sample of Caucasian youths with OW/OB. A recent meta-analysis demonstrated that in individuals without diabetes, HbA1c values were higher in Blacks, Asians and Latinos when compared to white individuals [21]. Despite the different ethnicity of our sample, we confirm the better performance of HbA1c ≥ 5.5%, in agreement with the multi-ethnic studies by Novicka et al. [7] and Tsay et al. [19], and the Chinese study by Poon et al. [10]. In this context, it is interesting to note that the HbA1c value ≥ 5.5% corresponds to the 95th percentile reported by Hovestadt et al. in 2455 young Germans (age 0.5–18 years) of whom 76.5% were normal weight [22].

The association between HbA1c and 1-h glucose is noteworthy. In fact, only one study performed in a small sample of 106 Turkish youths with OW/OB (7–18 years) demonstrated an association between G_60_ ≥ 155 mg/dL (derived from adult population) and HbA1c [23]. Interestingly, the best cut-off of HbA1c for G_60_ ≥ 155 mg/dL was ≥5.5% [23]. Our study confirmed the same cut-off of HbA1c using a pediatric cut-point of G_60_ (≥133 mg/dL). The association between HbA1c and high G_60_ is metabolically interesting, since this early impaired response to OGTT has been strictly associated with a high risk of beta-cell dysfunction and progression toward diabetes [15].

Indeed, a previous study undertaken in Chinese students showed that the optimal HbA1c cut-off of 5.6% had better accuracy for determining the clustering of cardiometabolic risk factors (sensitivity 35.1%; specificity 72.2%) compared to the cut-off of 5.7% (sensitivity 20.2%; specificity 84.3%) [24].

Our study presents some limitations, such as the cross-sectional design, which does not allow inferences on the progression of prediabetes to diabetes. In addition, measures to estimate beta-cell function were derived from the OGTT. Lastly, the study is limited to Caucasian youths with OW/OB. However, these features may also represent a strength, since children with OW/OB are considered as the main category at risk of prediabetes/diabetes. Furthermore, the analysis of the association between HbA1c, parameters of beta-cell function and cardiometabolic risk factors may represent another strength of our study.

The decision to use a lower cut-off of HbA1c might entail potential harms of overdiagnosis of prediabetes and higher costs associated with further evaluation of glucose dysmetabolism. Nevertheless, the perceived risk of illness may reinforce the intrinsic motivation to adhere to a weight loss program based on lifestyle. Indeed, an improved BMI trajectory after prediabetes identification was documented in youths with OW or OB followed longitudinally in a large academic-affiliated primary care network [25]. Therefore. prediabetes screening may be beneficial beyond its intended goal of identification of glucose dysmetabolism.

## 5. Conclusions

HbA1c values ≥ 5.5% may be considered for further evaluation of glucose dysmetabolism in youths with OW/OB. The higher sensitivity in detecting IFG, IGT and metabolic derangements compared to the cut-off of 5.7% may provide an opportunity to include a larger number of youths at risk of prediabetes in tighter management and follow-up. Longitudinal studies on the role of HbA1c in predicting diabetes and reducing diabetes-related comorbidities in children and adolescents are needed.

## Figures and Tables

**Figure 1 ijerph-20-00928-f001:**
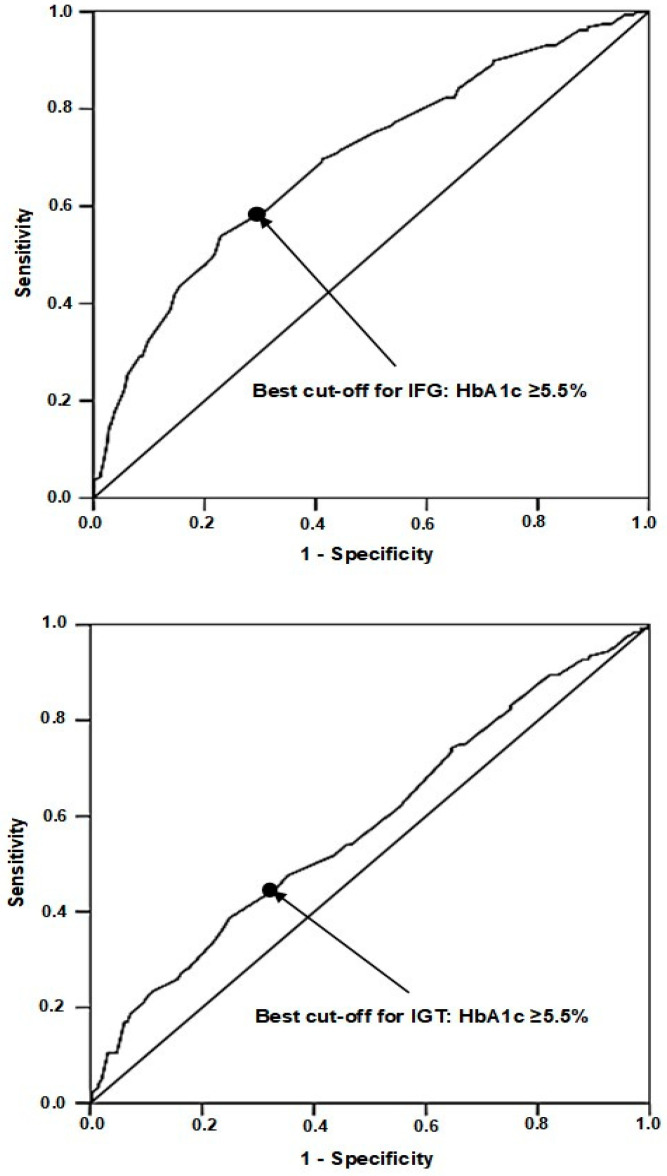
Area under curve of HbA1c with respect to IFG (**top** panel) and IGT (**bottom** panel).

**Table 1 ijerph-20-00928-t001:** Anthropometric, clinical, metabolic and beta-cell function variables according the two cut-offs of HbA1c.

	HbA1c < 5.5%	HbA1c ≥ 5.5%	*p* Value	HbA1c < 5.7%	HbA1c ≥ 5.7%	*p* Value
n = 1549	1045	504		1296	253	
Age (Years)	11.7 ± 2.7	11.6 ± 2.6	0.477	11.6 ± 2.6	11.7 ± 2.6	0.524
Girls, n (%)	521 (50)	254 (50)	0.842	645 (50)	130 (51)	0.638
BMI (kg/m^2^)	30.7 ± 5.4	31.4 ± 5.9	0.012	30.8 ± 5.5	31.4 ± 6.1	0.094
BMI-SDS	2.3 ± 0.6	2.4 ± 0.6	0.010	2.3 ± 0.6	2.4 ± 0.6	0.137
G_0_, (mg/dL)	87.1 ± 8.9	90.3 ± 10.7	<0.0001	87.4 ± 9.2	91.9 ± 11.0	<0.0001
G_60_, (mg/dL) *	122.4 ± 28.1	131.0 ± 28.5	<0.0001	123.7 ± 28.4	132.9 ± 27.9	<0.0001
G_120_, (mg/dL)	108.5 ± 19.6	115.6 ± 22.1	<0.0001	109.5 ± 20.1	117.5 ± 22.6	<0.0001
HOMA-IR	3.5 (2.3–5.1)	4.1 (2.9–6.4)	<0.0001	3.6 (2.4–5.2)	4.5 (3.0–6.8)	<0.0001
1/I_0_ (μU/mL)	0.06 (0.04–0.08)	0.05 (0.04–0.08)	<0.0001	0.06 (0.04–0.09)	0.05 (0.04–0.08)	<0.0001
Disposition Index **	0.15 (0.10–0.23)	0.13 (0.08–0.20)	0.063	0.15 (0.10–0.23)	0.13 (0.08–0.18)	0.064
Cholesterol, mg/dL	153.7 ± 29.6	156.2 ± 28.5	0.110	154.1 ± 29.6	156.3 ± 27.4	0.288
TG/HDL ratio	1.7 (1.2–2.3)	1.9 (1.4–2.6)	<0.0001	1.7 (1.3–2.4)	2.0 (1.5–2.8)	<0.0001
COL/HDL ratio	3.3 ± 0.9	3.5 ± 0.9	<0.0001	3.4 ± 0.9	3.6 ± 0.9	<0.0001
ALT (IU/mL)	21.0 (15.0–30.0)	24.0 (18.0–32.0)	<0.0001	21.0 (16.0–29.8)	25.0 (19.0–36.0)	<0.0001
Systolic BP (mmHg)	113.2 ± 14.0	114.2 ± 13.7	0.162	113.5 ± 13.9	113.7 ± 14.2	0.770
Diastolic BP (mmHg)	67.5 ± 9.3	68.2 ± 9.7	0.160	67.6 ± 9.4	68.3 ± 9.9	0.279

Data are expressed as mean ± standard deviation, median (IQ range), n (%). * n = 1430; ** n = 972. ALT, alanine aminotransferase; BMI, Body Mass Index; BMI–SDS, Body Mass Index–Standard Deviation Score; BP, blood pressure; COL/HDL, cholesterol/HDL-Cholesterol; G_0,_ fasting glucose; G_60,_ 1-h glucose; G_120,_ 2-h glucose; I_0,_ fasting insulin; TG/HDL-C, Triglycerides/HDL-Cholesterol.

**Table 2 ijerph-20-00928-t002:** Characteristics of youths with HbA1c < 5.7% reclassified by cut-off of 5.5%.

	HbA1c < 5.5%	HbA1c ≥ 5.5%	*p* Value
n = 1296	1045	251	
Age (Years)	11.7 ± 2.7	11.4 ± 2.6	0.149
Girls, n (%)	321 (49)	71 (47)	0.696
BMI (kg/m^2^)	30.7 ± 5.4	31.4 ± 5.7	0.057
BMI-SDS	2.3 ± 0.6	2.4 ± 0.6	0.034
G_0_, (mg/dL)	87.1 ± 8.9	88.7 ± 10.2	0.011
G_60_, (mg/dL) *	122.4 ± 28.1	129.0 ± 29.0	0.002
G_120_, (mg/dL)	108.5 ± 19.6	113.8 ± 21.4	<0.0001
HOMA-IR	3.5 (2.3–5.1)	3.9 (2.7–5.6)	<0.0001
1/I_0_ (μU/mL)	0.06 (0.04–0.08)	0.05 (0.04–0.08)	<0.0001
Disposition Index **	0.15 (0.10–0.23)	0.12 (0.09–0.16)	0.387
Cholesterol, mg/dL	153.7 ± 29.6	156.0 ± 29.0	0.228
TG/HDL ratio	1.7 (1.2–2.3)	1.9 (1.4–2.5)	0.002
COL/HDL ratio	3.3 ± 0.9	3.5 ± 0.9	0.042
ALT (IU/mL)	21.0 (15.0–30.0)	22.0 (17.0–29.0)	0.154
Systolic BP (mmHg)	113.2 ± 14.0	114.7 ± 13.2	0.115
Diastolic BP (mmHg)	67.5 ± 9.3	68.1 ± 9.6	0.348

Data are expressed as mean ± standard deviation, median (IQ range), n (%). * n = 1186; ** n = 809. ALT, alanine aminotransferase; BMI, Body Mass Index; BMI–SDS, Body Mass Index–Standard Deviation Score; BP, blood pressure; COL/HDL, cholesterol/HDL-Cholesterol; G_0,_ fasting glucose; G_60,_ 1-h glucose; G_120,_ 2-h glucose; I_0,_ fasting insulin; TG/HDL-C, Triglycerides/HDL-Cholesterol.

**Table 3 ijerph-20-00928-t003:** Performance of HbA1c ≥ 5.5% or HbA1c ≥ 5.7% in relation to parameters of glucose dysmetabolism, beta-cell dysfunction or cardiometabolic risk factors.

Factors	AUC	Sensitivity	Specificity
IFG			
HbA1c ≥ 5.5%	0.64 (0.60–0.69)	0.58 (0.56–0.61)	0.70 (0.68–0.73)
HbA1c ≥ 5.7%	0.62 (0.57–0.67)	0.39 (0.36–0.41)	0.86 (0.84–0.88)
IGT			
HbA1c ≥ 5.5%	0.56 (0.51–0.61)	0.44 (0.41–0.46)	0.68 (0.66–0.71)
HbA1c ≥ 5.7%	0.55 (0.50–0.61)	0.26 (0.24–0.28)	0.85 (0.83–0.86)
G_60_ ≥ 133 mg/dL			
HbA1c ≥ 5.5%	0.58 (0.55–0.61)	0.43 (0.40–0.46)	0.72 (0.70–0.75)
HbA1c ≥ 5.7%	0.55 (0.52–0.58)	0.24 (0.21–0.26)	0.87 (0.85–0.88)
Insulin-resistance			
HbA1c ≥ 5.5%	0.56 (0.53–0.59)	0.37 (0.34–0.39)	0.75 (0.73–0.77)
HbA1c ≥ 5.7%	0.54 (0.51–0.57)	0.19 (0.17–0.21)	0.89 (0.87–0.90)
Low insulin-sensitivity			
HbA1c ≥ 5.5%	0.55 (0.51–0.58)	0.39 (0.37–0.42)	0.70 (0.67–0.72)
HbA1c ≥ 5.7%	0.54 (0.51–0.57)	0.22 (0.20–0.24)	0.86 (0.84–0.87)
Low disposition index			
HbA1c ≥ 5.5%	0.55 (0.51–0.59)	0.40 (0.37–0.43)	0.70 (0.67–0.73)
HbA1c ≥ 5.7%	0.52 (0.48–0.57)	0.20 (0.18–0.23)	0.84 (0.82–0.87)
High TG/HDL ratio			
HbA1c ≥ 5.5%	0.55 (0.51–0.58)	0.40 (0.37–0.42)	0.70 (0.68–0.72)
HbA1c ≥ 5.7%	0.55 (0.51–0.58)	0.23 (0.21–0.25)	0.86 (0.84–0.88)
High COL/HDL ratio			
HbA1c ≥ 5.5%	0.55 (0.51–0.58)	0.40 (0.37–0.42)	0.70 (0.67–0.72)
HbA1c ≥ 5.7%	0.53 (0.50–0.57	0.21 (0.19–0.23)	0.85 (0.83–0.87)
High ALT			
HbA1c ≥ 5.5%	0.54 (0.51–0.57)	0.37 (0.35–0.39)	0.71 (0.68–0.73)
HbA1c ≥ 5.7%	0.54 (0.51–0.57)	0.21 (0.19–0.23)	0.87 (0.85–0.89)

G_60,_ 1-h glucose; TG/HDL-C, Triglycerides/HDL-Cholesterol; COL/HDL, cholesterol/HDL-Cholesterol; ALT, alanine aminotransferase.

## Data Availability

Not applicable.

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
