# Peer review of "Which Is the Most Appropriate Cut-Off of HbA1c for Prediabetes Screening in Caucasian Youths with Overweight or Obesity?"

_ijerph, 2023, doi:10.3390/ijerph20020928_

Round 1

Reviewer 1 Report

This manuscript is of seemingly simple logic but with not clear explanations. The authors raised definite and prioritized challenge of the proper diagnosis of diabetes mellitus and prediabetic condition in adolescents and child. Overall, the authors made a comprehensive workflow by collecting a large dataset. Such sufficient sample size permits to conduct quiet string statistical analysis. However, I would expect much stronger regressive models and tests considering that authors collected data of more than 1,500 subjects.

It should also be notice that reframing of indicators or cut-off thresholds actual for the current clinical levels are fraught with heavier socio-economic burden and responsibilities, so reconsideration of indicators and thresholds should follow the cost-of-illness assay.

Some not critical issues still should be corrected before the manuscript cam be accepted:

1.       Introduction: HbA1c is a glycated hemoglobin but not glycosylated.

2.       Material and Methods: I suggest, the last revision of WMA Declaration of Helsinki was in 2013 (revision Fortaleza), not in 1983. Please, check it.

3.       Figure 1: why not to show ROC in respect of IGT on the same graph to fit the best cut-off level of HbA1c?

4.       The AUC values in respect of both IFG and IGT are rather poor and made only 0.695 and 0.581 albeit higher values are expected because the sample size (dimensionality) od the study population is sufficient. I suggest this is because of weal segmentation of the study population.

5.       Why did HbA1c ≥5.5% demonstrate lower specificity compared to HbA1c ≥5.7%. Is it because of higher false-positive results caused by non-diabetic conditions? For example, high HbA1c can be observed in patients with iron-deficiency anemia whereas, in oppose, low values of HbA1c at hemolytic anemia. Could authors explain or evaluate a possible input of non-diabetic conditions in specificity variable?

6.       I think, comparison between >5.5% vs. 5.7% subgroups would be more valuable and might underline tenets targeted by the authors more proper.

7.       Was it possible to stratify the study cohort based on another indicator instead of HbA1c to provide cross-validity of proposed suggestions? Because studied groups of >5.7 and >5.5 are out of proportion obviously because the group with >5.5. includes all subjects with >5.7. In this respect, Figure 2, for example, reflects putative distribution of phenotypes proportion because grey bars (>5.5) accumulate data taken from the >5.7 group.

Author Response

We thank the reviewer for his/her suggestions, that allowed us to improve the paper.

Regarding the issue cost/benefit, the following sentence was added in the Discussion.

The decision to use a lower cut off of HbA1c might entail potential harms of overdiagnosis of prediabetes and higher costs associated with further evaluation of glucose dysmetabolism. Nevertheless, the perceived risk of illness may reinforce the intrinsic motivation to adhere to a weight loss treatment  based on lifestyle. Indeed, an improved BMI trajectory after prediabetes identification was documented  in youths with OW or OB followed longitudinally in a large academic-affiliated primary care network. (25). Therefore. prediabetes screening may be beneficial beyond its intended goal of identification of glucose dysmetabolism.

Some not critical issues still should be corrected before the manuscript cam be accepted:

  1. Introduction: HbA1c is a glycated hemoglobin but not glycosylated.

amended

  1. Material and Methods: I suggest, the last revision of WMA Declaration of Helsinki was in 2013 (revision Fortaleza), not in 1983. Please, check it.

amended

  1. Figure 1: why not to show ROC in respect of IGT on the same graph to fit the best cut-off level of HbA1c?

A new graph showing ROC  with respect to IGT was added

  1. The AUC values in respect of both IFG and IGT are rather poor and made only 0.695 and 0.581 albeit higher values are expected because the sample size (dimensionality) of the study population is sufficient. I suggest this is because of weal segmentation of the study population.

Yes, this may be the explanation.

  1. Why did HbA1c ≥5.5% demonstrate lower specificity compared to HbA1c ≥5.7%. Is it because of higher false-positive results caused by non-diabetic conditions? For example, high HbA1c can be observed in patients with iron-deficiency anemia whereas, in oppose, low values of HbA1c at hemolytic anemia. Could authors explain or evaluate a possible input of non-diabetic conditions in specificity variable?

We thank the reviewer for this observation that led us to improve the Methods section. The individuals who were included in the study did not show any other health condition apart obesity. The following sentence was added: “Other exclusion criteria were genetic causes of obesity, systemic and endocrine diseases, and use of medications affecting glucose metabolism”.

  1. I think, comparison between >5.5% vs. 5.7% subgroups would be more valuable and might underline tenets targeted by the authors more proper.

We compared the two groups and we did not found any significant differences. This sentence was added: No significant differences were found by comparing groups with HbA1c >5.5% to HbA1c >5.7%.  

  1. Was it possible to stratify the study cohort based on another indicator instead of HbA1c to provide cross-validity of proposed suggestions? Because studied groups of >5.7 and >5.5 are out of proportion obviously because the group with >5.5. includes all subjects with >5.7. In this respect, Figure 2, for example, reflects putative distribution of phenotypes proportion because grey bars (>5.5) accumulate data taken from the >5.7 group.

Yes, we agree with your comment and the two figures have been withdrawn.

In order to provide cross validity of the HbA1c 5.5% cut-off, we made new calculations by comparing youths reclassified by HbA1c levels below or above 5.5%, after excluding the 253 individuals with HbA1c ≥ 5.7%. A similar procedure was adopted for another successful paper (Bonito PD, et al The American Academy of Pediatrics hypertension guidelines identify obese youth at high cardiovascular risk among individuals non-hypertensive by the European Society of Hypertension guidelines. Eur J Prev Cardiol. 2020 Jan;27(1):8-15. ).With the only exception of ALT, this subanalysis confirmed that significantly higher values of G0, G60, G120, HOMA-IR, TG/HDL-C, COL/HDL-C, and a lower value IS were found in youths with HbA1c >5.5<5.7% compared to <5.5%, as it occurred in the initial sample that included also youths with HbA1c ≥5.7% . An additional table was added (Table 2), that we feel is comprehensive and at the same time synthetic for readers.

Reviewer 2 Report

Well-written manuscript on an issue of clinical importance. Could be strengthened by discussion of the implications of implementing a lower HbA1c cut-off in this population – there would be a cost associated with the further evaluation of glucose dysmetabolism for these additional individuals suggested in the conclusion, as well as costs associated with the management of additional individuals identified to have pre-diabetes, but it is also certainly arguable that the benefit of identifying these at-risk individuals would justify the costs.

Minor comment:

·         Line 58-60 – reference required for increased prevalence of overweight/obesity

Author Response

Well-written manuscript on an issue of clinical importance. Could be strengthened by discussion of the implications of implementing a lower HbA1c cut-off in this population – there would be a cost associated with the further evaluation of glucose dysmetabolism for these additional individuals suggested in the conclusion, as well as costs associated with the management of additional individuals identified to have pre-diabetes, but it is also certainly arguable that the benefit of identifying these at-risk individuals would justify the costs.

We thank the reviewer for this suggestion. The following sentence was added in the Discussion.

The decision to use a lower cut off of HbA1c might entail potential harms of overdiagnosis of prediabetes and higher costs associated with further evaluation of glucose dysmetabolism. Nevertheless, the perceived risk of illness may reinforce the intrinsic motivation to adhere to a weight loss treatment  based on lifestyle. Indeed,  an improved BMI trajectory after prediabetes identification was documented  in youths with OW or OB followed longitudinally in a large academic-affiliated primary care network. (Vajravelu ME, Lee JM, Shah R, Shults J, Amaral S, Kelly A. Association between prediabetes diagnosis and body mass index trajectory of overweight and obese adolescents. Pediatr Diabetes. 2020 Aug;21(5):743-746. doi: 10.1111/pedi.13028). Therefore. prediabetes screening may be beneficial beyond its intended goal of identification of glucose dysmetabolism.

  • Line 58-60 – reference required for increased prevalence of overweight/obesity

The following reference was added: Skinner AC, Ravanbakht SN, Skelton JA, Perrin EM, Armstrong SC. Prevalence of Obesity and Severe Obesity in US Children, 1999-2016. Pediatrics. 2018 Mar;141(3):e20173459.  Erratum in: Pediatrics. 2018 Sep;142(3):.

Reviewer 3 Report

1. Please improve the quality of fig. 1. The text size should be increased to make it properly readable for others.

2. Add more recent references if available.

Author Response

  1. Please improve the quality of fig. 1. The text size should be increased to make it properly readable for others.

The text size has been increased. Note that a new panel was added to figure 1, as suggested by reviewer 1

  1. Add more recent references if available.

We cited all the relevant references inherent the topic and we did not find any updated reference about it.

The English text has been revised..